# Factors Correlating to the Development of Hepatitis C Virus Infection among Drug Users—Findings from a Systematic Review and Meta-Analysis

**DOI:** 10.3390/ijerph16132345

**Published:** 2019-07-02

**Authors:** Biao Zhou, Gao Feng Cai, Hua Kun Lv, Shuang Fei Xu, Zheng Ting Wang, Zheng Gang Jiang, Chong Gao Hu, Yong Di Chen

**Affiliations:** 1Department of Scientific Research and Information Management, Zhejiang Provincial Center for Disease Control and Prevention, 3399 Binsheng Road, Hangzhou 310051, China; 2Department of Immunization Planning, Zhejiang Provincial Key Laboratory of Infectious Disease Vaccine and Prevention and Control, 3399 Binsheng Road, Hangzhou 310051, China

**Keywords:** drug use, hepatitis C virus, risk factor, meta-analysis

## Abstract

Hepatitis C remains a significant public health threat. However, the main routes of transmission have changed since the early 1990s. Currently, drug use is the main source of hepatitis C virus (HCV) infection, and some measures have been successively implemented and additional studies have been published. However, the factors correlating with HCV infection failed to clearly define. Our study pooled the odds ratios (ORs) with 95% confidence intervals (CIs) and analyzed sensitivity by searching data in the PubMed, Elsevier, Springer, Wiley, and EBSCO databases. Publication bias was determined by Egger’s test. In our meta-analysis, HCV-infected and non-HCV-infected patients from 49 studies were analyzed. The pooled ORs with 95% CIs for study factors were as follows: Injecting drug use 10.11 (8.54, 11.97); sharing needles and syringes 2.24 (1.78, 2.83); duration of drug use >5 years 2.39 (1.54, 3.71); unemployment 1.50 (1.22, 1.85); commercial sexual behavior 1.00 (0.73, 1.38); married or cohabiting with a regular partner 0.88 (0.79, 0.98), and sexual behavior without a condom 1.72 (1.07, 2.78). This study found that drug users with histories of injecting drug use, sharing needles and syringes, drug use duration of >5 years, and unemployment, were at increased risk of HCV infection. Our findings indicate that sterile needles and syringes should be made available to ensure safe injection. In view of that, methadone maintenance treatment can reduce or put an end to risky drug-use behaviors, and should be scaled up further, thereby reducing HCV infection.

## 1. Introduction

Hepatitis C is a viral infectious disease caused by the HCV, which is characterized by diffuse liver damage [1]. In 2003, the World Health Organization (WHO) estimated that the global prevalence of HCV infection was about 3.0%, with about 3–4 million new infections every year, about 130–150 million chronic infections worldwide, and about 672,000 deaths annually from acute viral hepatitis C infections and hepatitis C-related liver cancer and cirrhosis [2]. At present, the new curative treatment for HCV infection, e.g., directly acting antiviral, was available and convinced in safety, and tolerability [3,4,5].

However, the main route of HCV infection has changed since the early 1990s. Before the 1990s, blood transfusions and the use of contaminated blood products were the main sources of HCV infection [2,3,4,5,6]. Whereas currently, drug use is the main source of HCV infection in most developed and developing countries [2], and statistics have shown that over 50% of drug users are intravenous drug users [7,8,9], with an estimated 12 million injecting drug users worldwide [10]. HCV infection rates have been increasing since 2006 in some areas, especially among the younger population [9]. In addition, in the last 10 years, the number of new drug users has been increasing and there are various ways of drug use [11,12,13,14]. Pan et al. also showed that the club drug users had a high prevalence of HCV infection, as well as people with high-frequency unprotected sexual behavior and less available intervention services [15]. These findings suggest that new drugs have become a new threat to human health.

In fact, in recent times, the number of global drug users has increased rapidly, from 185 million in 2004 to 250 million in 2015 [10,16]. Similarly, the number of registered drug users in China has risen rapidly, from 70,000 in 1990 to 3 million in 2015 [17]. In view of this significant population of drug users and the risk of cross infection with HCV among drug users, the impact of HCV infection among this population remains significant and constitutes a major health burden.

In recent years, some practical measures and strategies have been implemented to tackle the spread of infections among drug users, such as publicizing health information, making condoms available in public places, providing needle/syringe exchange centers, as well as opening methadone maintenance treatment clinics that help reduce risky drug-use behaviors thereby reducing the risk of transmission of blood-borne infections [8,18,19,20,21,22,23,24,25]. However, the rates of HCV infection remain high among drug users [25,26]. Many factors influence the development of HCV infection among drug users, such as injecting behavior, sharing needles and/or syringes, the duration of drug use, and high-risk sexual behavior [27,28,29,30,31,32,33,34,35].

In 2006, Xia et al. performed meta-analysis on data collected from drug users in China to determine factors that correlate with the occurrence of HCV infection [26], and Stone et al. performed another meta-analysis for incarceration history and risk of HCV acquisition among people who injected drugs in 2018 [36]. In another study, Hagan et al. collected data from published or released reports between January 1989 and December 2006 and data from scientific conferences between December 2006 April 2010, and meta-analyzed the effects of risk-reduction interventions [25]. They concluded that combined substance-use treatment and support for safe injection were the most effective interventions for reducing HCV seroconversion. These findings implied that appropriate intervention can prevent HCV infection among injection drug users. Studies have also demonstrated that the hazards posed by certain risk factors can be controlled. However, this meta-analysis was performed on a limited number of studies and limitations were imposed as a result of literature being unavailable. For example, in 2004, Lin et al. completed a quasi-experiment design in a controlled community intervention study that included a needle and syringe exchange program, peer education and health education, provision of free needles and syringes, and the collection of used needles, which was implemented for 10 months to injecting drug users in an intervention community, but no intervention measures were implemented in a control community for comparison [37]. A number of other studies investigating factors correlating to HCV infection in drug users have been published since 2006 [27,28,38,39,40,41,42,43,44,45,46,47,48,49,50,51,52,53,54,55,56,57,58,59,60,61,62,63,64,65,66,67,68,69,70,71,72,73,74,75,76,77]; however, the contribution of each of the factors identified in such studies remains unclear or in some cases is even contradictory.

## 2. Materials and Methods

### 2.1. Literature Search Strategy

Searches were performed in specified databases on the BoKu data service platform. We used the following search terms “Hepatitis C or HCV” and “drug use or drug addiction” in the search field “Title/Abstract,” and searched six international databases, namely PubMed, OVID, Springer, Wiley, Elsevier, and EBSCO. We also used the search terms “Hepatitis C or “HCV” and “drug use or drug addiction” in the search field “Abstract,” and searched the Chinese Medical Journal Database and Chinese National Knowledge Infrastructure. The searches were completed in the last week of March 2019.

### 2.2. Inclusion and Exclusion Criteria

The eligibility criteria for the studies included in this meta-analysis were: (1) The study is an original research; (2) the study was an observational study with specific temporal and geographic characteristics; (3) the study was published with the full text available; (4) all cases and controls were drug users and the source of research objects was clearly stated; (5) major influencing factors were reported; and (6) hepatitis C was diagnosed by the national diagnostic criteria that existed at that time [78].

Literature was excluded from the meta-analysis when: (1) Based on the data reported, the odds ratio (OR) with 95% confidence interval (CI) could not be achieved by calculating the major influencing factors; (2) the literature duplicated the same research; (3) according to the source of the research objects, the province (state) was used as the screening repeated research object analysis unit and for studies with the same or cross research objects, only one of the studies was included and the others were excluded; and (4) according to the declaration by Ebrahim et al., the literature satisfying the number of items in the corresponding research type declaration was less than half the total number of items [79,80].

### 2.3. Data Extraction

A pre-made form was used for data extraction. The literature was assessed one-by-one and the form was completed by two trained reviewers. The following data were extracted from the qualified studies: First author, year of the study, location, sample size, the number of drug users in the HCV-infected group and the non-HCV-infected group, the number of males and females or the male to female ratio, and age distribution among drug users.

Discrepancies between the assessment results acquired by the two reviewers were resolved by checking the original documents and discussing.

### 2.4. Sensitivity Analysis

In this meta-analysis, the studies with the maximum weight were omitted from the subgroup analysis. The remaining studies were pooled, and the pooled OR_weight_ values with 95% CIs for each study factor were obtained. The pooled OR_weight_ values were then compared with the pooled ORs before being omitted from the study.

### 2.5. Statistical Analysis

In this meta-analysis, the main indicators were the ORs with 95% CIs. Following a heterogeneity test, the fixed effects model was used to analyze factors without heterogeneity for the different studies and the random effect model was used to analyze factors with heterogeneity using the Review Manager 5.1 software. (Cochrane Collaboration, Rigshospitalet, Denmark). Heterogeneity was evaluated using Cochran’s chi-square test with a significance level of α = 0.1 and using I^2^ statistics with heterogeneity accepted as I^2^ ≤ 50% [81]. In this meta-analysis, I^2^ ≤ 50% was accepted. The Egger’s test was performed using the software Stata version 11.0 (Stata Corp., College Station, TX, USA), with a significance level of α = 0.05.

## 3. Results

### 3.1. Literature Search

Based on the inclusive criteria and exclusive criteria, all articles were retrieved and carefully reviewed to assess the eligibility. Forty-nine eligible studies were identified after a screening of 1109. The selection of studies for the meta-analysis is shown in Figure 1 [82].

### 3.2. Characteristics of the Studies

Among the forty-nine studies, the 10 study factors used to pool ORs with 95% CIs were as follows: Injecting drug use (43 studies, 53,860 cases, 69,747 controls); sharing needles and syringes (33 studies, 40,777 cases, 23,361 controls); duration of drug use >5 years (12 studies, 10,282 cases, 8,794 controls); unemployment (6 studies, 8,361 cases, 5,420 controls); sex (male) (39 studies, 57,403 cases, 72,922 controls); education level ≤9 years (29 studies, 46,931 cases, 61,841 controls); sexual behavior without a condom (15 studies, 10,032 cases, 24,156 controls); Han ethnic group (15 studies, 12,014 cases, 16,129 controls); married or cohabiting with a regular partner (26 studies, 46,626 cases, 62,107 controls); commercial sexual behavior (14 studies, 9,698 cases, 28,505 controls). Among the forty-nine studies, the proportion of 4 studies (injecting only) was 8.16%, the proportion of 2 studies (non-injecting) was 4.08%, and the proportion of 43 studies (injecting and non-injecting) was 87.76%.

The characteristics of all studies evaluated in this meta-analysis are shown in Table 1.

### 3.3. Results of Pooled ORs

In this meta-analysis, the pooled ORs with 95% CIs for study factors were as follows: Injecting drug use 10.11 (8.54, 11.97); sharing needles and syringes 2.24 (1.78, 2.83); duration of drug use >5 years 2.39 (1.54, 3.71); unemployment 1.50 (1.22, 1.85); commercial sexual behavior 1.00 (0.73, 1.38); sex (male) 1.04 (0.91, 1.18); married or cohabiting with a regular partner 0.88 (0.79, 0.98); Han ethnic group 0.94 (0.73, 1.20); sexual behavior without a condom 1.72 (1.07, 2.78); and education level ≤9 years 1.05 (0.92, 1.21).

The pooled ORs with their 95% CIs for study factors are detailed in Figure 2, Figure 3 and Figure 4, and the axes of the figures mean OR = 1.

### 3.4. Results of Heterogeneity Evaluation

A heterogeneity test showed that variations among studies for the pooled ORs with 95% CIs for factors including injecting drug use, sharing needles and syringes, drug use duration of >5 years, unemployment, commercial sexual behavior, sex (male), married or cohabiting with a regular partner, Han ethnic group, sexual behavior without a condom, and an education level of ≤9 years were statistically significant (*p* < 0.10). The effects of these factors were then pooled using the random effect model. These results are detailed in Figure 2, Figure 3 and Figure 4.

### 3.5. Publication Bias

In this meta-analysis, a funnel plot for the duration of drug use was symmetrical, with the axis of symmetry (OR = 1) being to the right of center, as detailed in Figure 5. The results of Egger’s test for study factors were all *p* > 0.05, as detailed in Table 2.

### 3.6. Sensitivity Analysis

In view of the reliability of the pooled ORs using the random effect model for terms including injecting drug use, drug use duration of >5 years, unemployment, commercial sexual behavior, sex (male), married or cohabiting with a regular partner, Han ethnic group, sharing needles and syringes, sexual behavior without a condom, and an education level of ≤9 years, we omitted studies with the highest weights, pooled the remaining studies, and acquired the OR_weight_ values with 95% CIs. These pooled values were compared with those obtained before the studies were omitted for qualitative and quantitative comparisons and no major changes in the pooled ORs with 95% CIs were observed for any of the study factors, as detailed in Table 2.

## 4. Discussion

This study found that drug users with a history of injecting drug use, and/or sharing needles/syringes, drug use duration of >5 years, and/or unemployment, and/or sexual behavior without a condom, were at increased risk of HCV infection, whereas drug users who were married or cohabiting with a regular partner were at decreased risk of developing HCV infection. This study also found that, for drug users (male), commercial sexual behavior, Han ethnicity, an education level of ≤9 years, did not affect the risk of developing HCV infection.

In general, exposure to HCV-contaminated needles and syringes increases the risk of HCV infection, and such exposure may be common among injecting drug users [83,84]. The findings of our meta-analysis confirmed that drug users with histories of injecting drug use were at increased risk of HCV infection, and this result was consistent with those of Xia et al. [26]. However, the findings of this meta-analysis that injecting drug users with a history of sharing needles/syringes were at increased risk of developing HCV infection was not consistent with the results reported in the meta-analysis by Xia et al. in 2008, and this may be related to the fact that only three studies on sharing needles were pooled in Xia et al.’s study and that this small sample size led to low test efficiency and unreliable results [26].

The findings of this meta-analysis also showed that drug use duration of >5 years was a risk factor for developing HCV infection, which may relate to the longer the duration of drug use, the greater the opportunity to be exposed to HCV-contaminated needles or goods, potentially leading to an infection. This result was consistent with the findings of a previous meta-analysis report [26].

In view of that, methadone maintenance treatment can reduce or put an end to risky drug-use behaviors. This discovery suggests methadone maintenance treatment should be scaled up further so as to shorten the duration of drug use and reduce the risk of HCV infections [24].

The findings of this meta-analysis also showed that sexual behavior without a condom was a risk factor for developing HCV infection. The results of the meta-analysis by Xia et al. in 2008′s meta-analysis reported that high-risk sexual practices were strongly associated with injecting drug behavior [26], but the magnitude of high-risk sexual behavior or the correlation between high-risk sexual behavior and drug-injecting behavior and their contribution to the occurrence of HCV infection could not be determined, thus requiring further study. In our study, there was a high proportion of injecting drug behavior and high-risk sexual behavior among the drug users, but related information on individual cases was not available.

The use of amphetamine-type stimulants is currently on the rise, as is unprotected sexual behavior becoming more common and leads to a high prevalence of HCV infection among the club drug user [17,85,86,87], strategies therefore need to be implemented to try to reduce such behaviors and to help to reduce the progression of HCV infection. In addition, further meta-analyses for club drug use will be done when there is enough literature.

The findings of this meta-analysis also showed that unemployment among drug users increased the risk of developing HCV infection, and this finding was consistent with the results of meta-analysis of human immunodeficiency virus (HIV) infection, which has a similar transmission route, among drug users [88].

The findings of our meta-analysis showed that drug users who were married or cohabiting with a regular partner were at decreased risk of developing HCV infection, and this finding was consistent with the results of meta-analysis published by Hagan et al. [27], and this may be related to the fact that these drug users had fewer sexual partners and fewer opportunities to be exposed to HCV-infected bodily fluids. However, the findings of this meta-analysis also showed that drug users with commercial sexual behavior, namely having multiple sexual partners, were not at increased risk of developing HCV infection, and this may be related to using club drug use with a shorter duration of drug use [15], although this requires further investigation.

The findings of this meta-analysis also showed that those of Han ethnicity, compared with those of other minority ethnic groups, were not at increased or decreased risk of developing HCV infection, and this finding was consistent with the results of meta-analysis published by Xia et al. [26]. Our findings also showed that drug users with an education level of ≤9 years were not at increased or decreased risk of developing HCV infection; however, this finding was inconsistent with that of a previous meta-analysis of HIV infection, which has a similar transmission route among drug users [88].

The limitations of this study were that even though the ORs of the study factors were pooled using a random–effect method, heterogeneity among studies might have influenced the findings. In addition, some study factors, for example, some racial classifications (white or black) were not available to be pooled. Lastly, few studies could be unavailable because of language limitations, in view of that, this meta-analysis’ publication bias was not statistically significant, and thus, this aspect influenced findings slightly.

## 5. Conclusions

This study found that drug users with histories of injecting drug use, sharing needles and syringes, drug use duration of >5 years, and unemployment, were confirmed to be at increased risk of HCV infection. Our findings indicate that high-risk drug users should be closely monitored and sterile needles and syringes should be made available to ensure safe injection. In view of that, methadone maintenance treatment can reduce or put an end to risky drug-use behaviors and should be scaled up further so as to shorten the duration of drug use, thereby reducing HCV infection.

## Figures and Tables

**Figure 1 ijerph-16-02345-f001:**
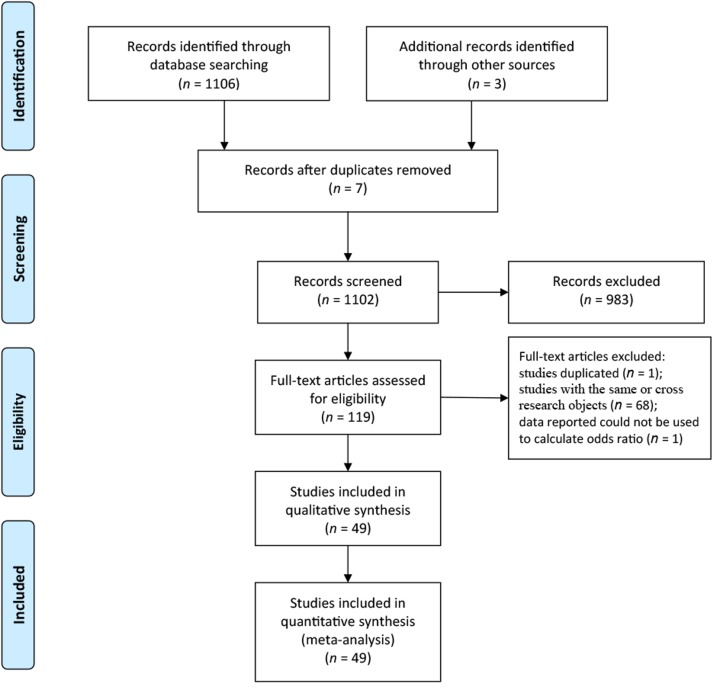
A flow chart of the studies selection process.

**Figure 2 ijerph-16-02345-f002:**
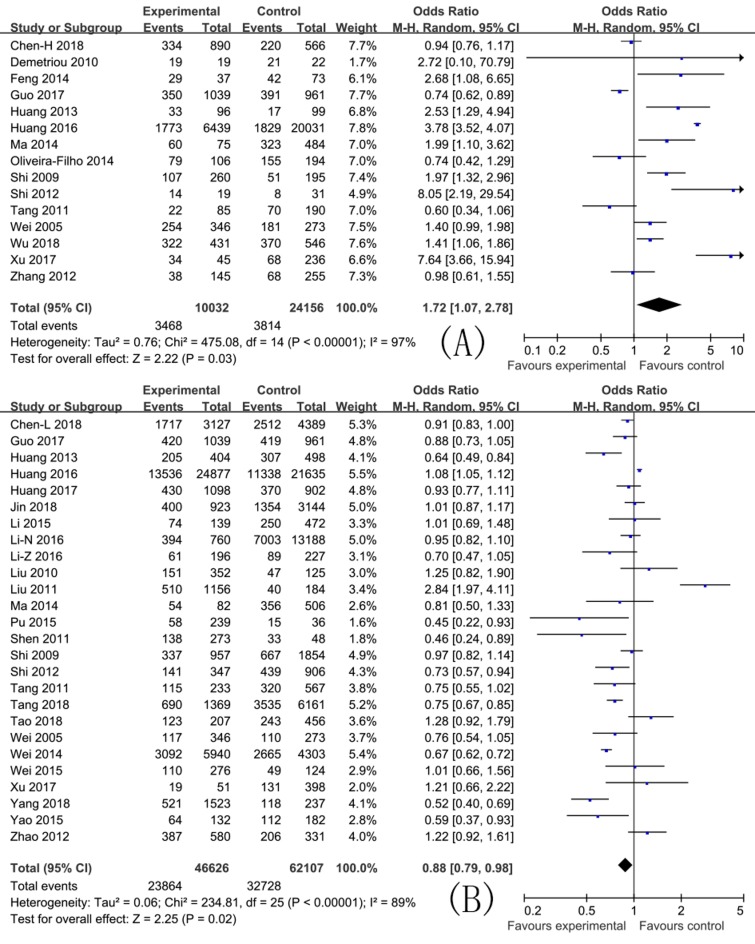
Effects of pooled ORs for factors correlating to the development of HCV infection among drug users ((**A**) sexual behavior without a condom; and (**B**) married or cohabiting with a regular partner).

**Figure 3 ijerph-16-02345-f003:**
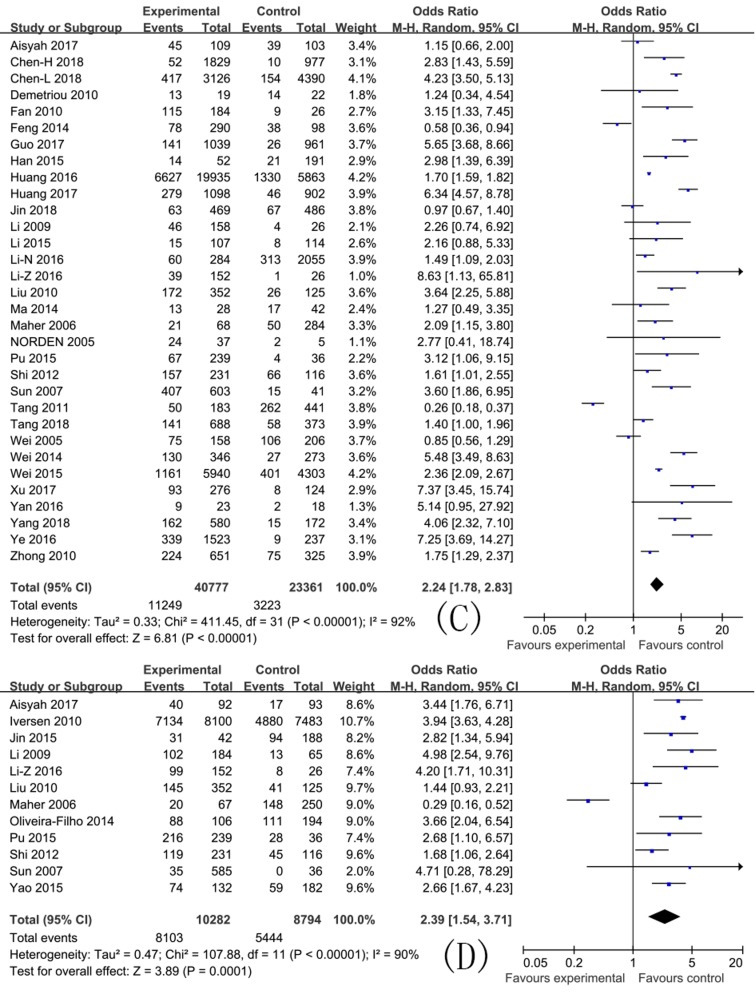
Effects of pooled ORs for factors correlating to the development of HCV infection among drug users ((**C**) Sharing needles and syringes; and (**D**) Duration of drug use >5 years).

**Figure 4 ijerph-16-02345-f004:**
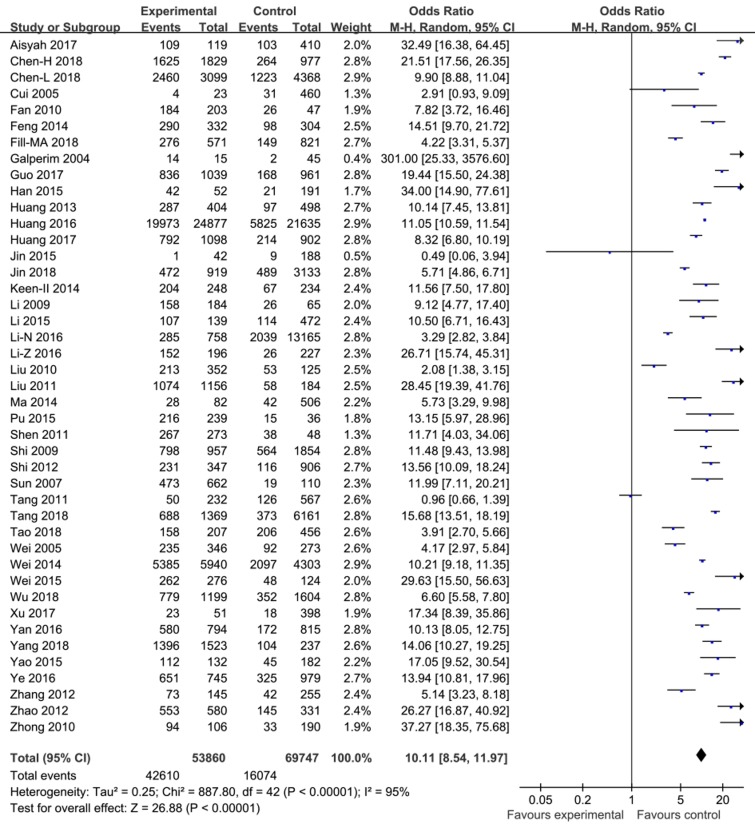
Effects of pooled ORs for injecting drug use correlating to the development of HCV infection among drug users.

**Figure 5 ijerph-16-02345-f005:**
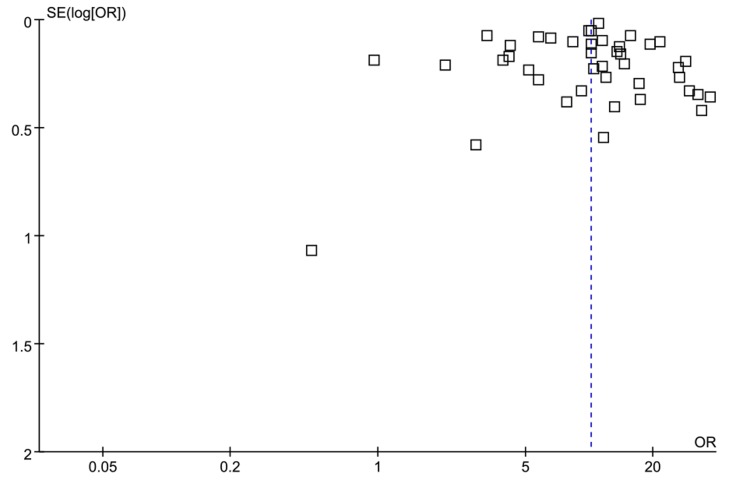
A funnel plot of the articles publication for the duration of drug use.

**Table 1 ijerph-16-02345-t001:** Characteristics of the studies.

Reference Number	Author	Year of Publication	Regions	Type of Drug Use	Participants Category (Case/Control)	Sample Size (Case/Control)	Male/Female	Age (Years) *
[38]	Chen Hua	2018	Sichuan, Mianyang	injecting and non-injecting	HCV-infected drug uses/non-HCV-infected drug uses	1829; 977	2016; 790	39.0 ± 7. 5
[39]	Wu Zhen Xiang	2018	Shanghai, Baoshan	injecting and non-injecting	HCV-infected drug uses/non-HCV-infected drug uses	1199; 1604	2138; 665	39.7 ± 9.86
[40]	Jin Jie	2018	Zhejiang, Hangzhou	injecting and non-injecting	HCV-infected drug uses/non-HCV-infected drug uses	923; 3144	3329; 638	36.33 ± 8.98
[41]	Xu Wen Xin	2017	Zhejiang, Jiaxing	injecting and non-injecting	HCV-infected drug uses/non-HCV-infected drug uses	51; 398	356; 93	27.50 ± 12.28
[42]	Ye.y	2016	Xinjiang, Wulumuji	injecting and non-injecting	HCV-infected drug uses/non-HCV-infected drug uses	745; 979	1 679; 49	35–45
[35]	Yun Chang Yan	2016	Yunnan, Haike	injecting and non-injecting	HCV-infected drug uses/non-HCV-infected drug uses	823; 786	-	33.8 ± 4.8
[43]	Li Ze	2016	Yunnan, Dali	injecting and non-injecting	HCV-infected drug uses/non-HCV-infected drug uses	196; 227	400; 23	15–62
[44]	Tao Yi Xin	2018	Qinghai, Xining	injecting and non-injecting	HCV-infected drug uses/non-HCV-infected drug uses	207; 456	401; 262	>20
[45]	Zhang Tao	2012	Zhejiang, Jinhua	injecting and non-injecting	HCV-infected drug uses/non-HCV-infected drug uses	145; 255	331; 69	31.61 ± 6.80
[46]	Shen Han Ding	2011	Yunnan, Jinning	injecting and non-injecting	HCV-infected drug uses/non-HCV-infected drug uses	273; 48	290; 31	20–78
[47]	Liu Qun	2011	Hubei, Wuhan	injecting and non-injecting	HCV-infected drug uses/non-HCV-infected drug uses	1156; 184	1000; 340	32.5 ± 6.2
[30]	Bruno Galperim	2004	Porto Alegre, RS, Brazil	injecting and non-injecting	HCV-infected drug uses/non-HCV-infected drug uses	15; 45	50; 10	31 ± 7
[31]	Lisa Maher	2006	Sydney, Australia	injecting only	HCV-infected drug uses/non-HCV-infected drug uses	68; 300	140; 228	>15
[48]	Aldemir B. Oliveira-Filho	2014	Pará, Brazil	non-injecting	HCV-infected drug uses/non-HCV-infected drug uses	106; 194	191; 109	32.5 ± 10.3
[49]	Wei Xiaoli	2014	Shanxi, Xian	injecting and non-injecting	HCV-infected drug uses/non-HCV-infected drug uses	5940; 4303	8653; 1590	37.4 ± 6.7
[50]	M. Zeremski	2012	New York, USA	non-injecting	HCV-infected drug uses/non-HCV-infected drug uses	11; 46	48; 9	44 ± 7
[51]	Larry Keen II	2014	Florida, USA	injecting and non-injecting	HCV-infected drug uses/non-HCV-infected drug uses	274; 208	284; 198	32.66 ± 7.01
[52]	Jenny Iversen	2010	New South Wales, Australia	injecting only	HCV-infected drug uses/non-HCV-infected drug uses	8100; 7483	10162; 5421	31 ± 8.8
[53]	Victoria L. Demetriou	2010	Nicosia, Cyprus	injecting only	HCV-infected drug uses/non-HCV-infected drug uses	19; 22	35; 6	27 (25–31)
[27]	Fill MA	2018	Tennessee, USA	injecting and non-injecting	HCV-infected drug uses/non-HCV-infected drug uses	571; 821	66:100	>18
[28]	D. N. Aisyah	2017	London, UK	injecting and non-injecting	HCV-infected drug uses/non-HCV-infected drug uses	119; 422	1093; 110	>18
[29]	Lillebil Norden	2005	Huddinge, Sweden	injecting only	HCV-infected drug uses/non-HCV-infected drug uses	37; 5	28; 14	-
[74]	Huang Dong Sheng	2013	Yunnan, Baoshan	injecting and non-injecting	HCV-infected drug uses/non-HCV-infected drug uses	404; 498	874; 28	-
[75]	Zhao Hong	2012	Neimenggu, Wuhai	injecting and non-injecting	HCV-infected drug uses/non-HCV-infected drug uses	580; 331	856; 55	18–63
[35]	Cui Xiu Ling	2005	Shanxi, Baoji	injecting and non-injecting	HCV-infected drug uses/non-HCV-infected drug uses	23; 460	427; 56	19–52
[76]	Shi Wen Ya	2012	Beijing, Fengtai	injecting and non-injecting	HCV-infected drug uses/non-HCV-infected drug uses	347; 906	954; 299	-
[77]	Zhong Hai Rong	2010	Jiangxi, Ganzhou	injecting and non-injecting	HCV-infected drug uses/non-HCV-infected drug uses	106; 190	237; 59	16–51
[32]	Wei Da Yin	2005	Sichuan, Liangshan	injecting and non-injecting	HCV-infected drug uses/non-HCV-infected drug uses	346; 273	519; 100	28.9 ± 6.4
[33]	Sun Yan	2007	Hunan, Changsha	injecting and non-injecting	HCV-infected drug uses/non-HCV-infected drug uses	662; 110	452; 320	15–53
[54]	Shi Ping	2009	Jiangsu, Nanjing	injecting and non-injecting	HCV-infected drug uses/non-HCV-infected drug uses	957; 1854	2305; 506	18–74
[55]	Fan Lin Jun	2010	Guangxi, Pingnan	injecting and non-injecting	HCV-infected drug uses/non-HCV-infected drug uses	203; 47	245; 5	37 (15–68)
[56]	Tang Xue Qin	2011	Jiangxi, Nanchang	injecting and non-injecting	HCV-infected drug uses/non-HCV-infected drug uses	233; 567	768; 32	18–77
[57]	Liu Hui Bin	2010	Shanxi, Yulin	injecting and non-injecting	HCV-infected drug uses/non-HCV-infected drug uses	352; 125	440; 37	20–52
[58]	Li Guang Qing	2009	Hunan, Bingzhou	injecting and non-injecting	HCV-infected drug uses/non-HCV-infected drug uses	184; 65	185; 64	32.32 (17–54)
[59]	Huang Dao Ping	2017	Hunan, Changde	injecting and non-injecting	HCV-infected drug uses/non-HCV-infected drug uses	1098; 902	1868; 132	33 (16–63)
[60]	Yang Kai	2018	Hubei, Yichang	injecting and non-injecting	HCV-infected drug uses/non-HCV-infected drug uses	1523; 288	1411; 400	44.78 ± 6.91
[61]	Chen Liang	2018	Fujian, Fuqing	injecting and non-injecting	HCV-infected drug uses/non-HCV-infected drug uses	3124; 4392	6630; 886	35.37 ± 8.97
[62]	Tang Ren Hai	2018	Yunnan, Dehong	injecting and non-injecting	HCV-infected drug uses/non-HCV-infected drug uses	1369; 6161	7176; 354	35.14 ± 10.9
[63]	Guo Yan	2017	Tianjiin, China	injecting and non-injecting	HCV-infected drug uses/non-HCV-infected drug uses	1039; 961	1642; 358	33 (34.5 ± 8.69)
[64]	Li Nin	2016	Henan, China	injecting and non-injecting	HCV-infected drug uses/non-HCV-infected drug uses	760; 13,195	11,224; 2724	37.32 ± 8.43
[65]	Huang Xi Ming	2016	Guangdong, China	injecting and non-injecting	HCV-infected drug uses/non-HCV-infected drug uses	24,877; 21,652	43,108; 3421	-
[66]	Yao Zhong Zheng	2015	Xinjiang, Wushi	injecting and non-injecting	HCV-infected drug uses/non-HCV-infected drug uses	132; 182	229; 15	19–69
[67]	Wei Li	2015	Guangxi, Liuzhou	injecting and non-injecting	HCV-infected drug uses/non-HCV-infected drug uses	276; 124	296; 104	-
[68]	Jin Hui Ya	2015	Gansu, Lanzhou	injecting and non-injecting	HCV-infected drug uses/non-HCV-infected drug uses	41; 189	120; 110	39.7–9.1
[69]	Ma Ji Xiong	2014	Gansu, Baiying	injecting and non-injecting	HCV-infected drug uses/non-HCV-infected drug uses	82; 506	548; 40	30.06 ± 6.3
[70]	Pu Li Fang	2015	Yunnan, Kaiyuan	injecting and non-injecting	HCV-infected drug uses/non-HCV-infected drug uses	239; 36	209; 66	41.6 ± 6.0
[71]	Li Feng	2015	Beeijing, Changping	injecting and non-injecting	HCV-infected drug uses/non-HCV-infected drug uses	139; 472	504; 107	>20
[72]	Han Xia	2014	Neimenggu, Huhehaote	injecting and non-injecting	HCV-infected drug uses/non-HCV-infected drug uses	52; 191	243; 0	>20
[73]	Feng Yan Jie	2014	Hebei, Qinhuangdao	injecting and non-injecting	HCV-infected drug uses/non-HCV-infected drug uses	332; 304	577; 59	>20

Note: *: mean ± standard deviation; mean (minimum–maximum); minimum–maximum; mean.

**Table 2 ijerph-16-02345-t002:** The subgroup characteristics of the study factors associated with HCV infection among drug users after omitting the studies with the maximum weight value for the ORs in the subgroup analysis and the results of Egger’s test.

Subgroup Analyses by Study Factors (1) *	Pooled OR with 95% CI before Reference Omitted (2)	Pooled OR with 95% CI after Reference Omitted (3)	Qualitative Comparison: Reversal of Pooled OR with 95% CI ((2) and (3) Compared)	Quantitative Comparison: Similar Values of Pooled OR with 95% CI ((2) and (3) Compared)	Reference Omitted	Egger’s Test	
						*t*	*p*-Value
Education level ≤9 years	1.05 (0.92, 1.21)	1.05 (0.91, 1.21)	No	Yes	[49]	−0.77	0.450
Sexual behavior without a condom	1.72 (1.07, 2.78)	1.50 (1.10, 2.03)	No	Yes	[65]	−1.79	0.097
Sharing needles and syringes	2.244 (1.78, 2.83)	2.31 (1.66, 3.23)	No	Yes	[65,67]	0.86	0.395
Han ethnic group	0.94 (0.73, 1.20)	0.96 (0.70, 1.30)	No	Yes	[62]	0.27	0.788
Married or cohabiting with a regular partner	0.88 (0.79, 0.98)	0.87 (0.78, 0.97)	No	Yes	[65]	−1.53	0.139
Sex (male)	1.04 (0.91, 1.18)	1.02 (0.90, 1.15)	No	Yes	[52,65]	−1.01	0.319
Commercial sexual behavior	1.00 (0.73, 1.38)	0.95 (0.61, 1.47)	No	Yes	[39,40,64]	−0.79	0.446
Unemployment	1.50 (1.22, 1.85)	1.48 (1.07, 2.06)	No	Yes	[49]	−0.23	0.831
Duration of drug use >5 years	3.49 (3.24, 3.75)	3.47 (3.22, 3.74)	No	Yes	[58]	−1.78	0.106
Injecting drug use	10.11 (8.54, 11.97)	10.21 (8.03, 12.97)	No	Yes	[49,61,65]	−0.35	0.731

Note: *: (1)—means Subgroup Analyses by Study Factors; (2)—means Pooled OR with 95% CI before Reference Omitted; (3)—means Pooled OR with 95% CI after Reference Omitted.

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
