# Peer review of "Factors Correlating to the Development of Hepatitis C Virus Infection among Drug Users—Findings from a Systematic Review and Meta-Analysis"

_ijerph, 2019, doi:10.3390/ijerph16132345_

Round 1

Reviewer 1 Report

Many thanks on addressing the previous comments.

There are numerous grammatical and spelling errors, especially in the revised sections of the manuscript. Once these errors are corrected I would be happy to endorse proceding to publication. 

Reviewer 2 Report

The reviewers have responded satisfactorily - but the manuscript overall still needs some English editing.

This manuscript is a resubmission of an earlier submission. The following is a list of the peer review reports and author responses from that submission.

Round 1

Reviewer 1 Report

Summary, strengths, and limitations.

The authors provide a concise yet broad summary of the literature concerning the currently identified major factors that correlate to the development of HCV infection in injecting drug users. The meta-review is more comprehensive than previous analyses and therefore provides more robust findings. However, the study could be improved by making the description of the original research papers (and analysis inclusion criteria) more thorough. The figure legends require much more detail and explanation, and the authors should take care to revise their manuscript carefully before resubmission because there are many sections that are incomplete or irrelevant. These limitations are elaborated in the points below.

Specific points that require addressing (please indicate the line number, and highlight the text, in the modified manuscript wherever new statements are inserted or other modifications are made)

·      As mentioned in your introduction, there is increased use of non-injectable illicit drugs, and it is therefore important to include somewhere in the manuscript whether or not the inclusion criteria for your meta analysis included all types of drug users or if this was limited to only injecting drug users.

·      If all types of drug users were included in your analysis, or there were mixed types of drug users included in the original research publications you have compiled in the table, please indicate what type of drugs were used by the drug users of each study in the table (e.g. add a new column called ‘type of drug use’ and mention for each study whether this was ‘injecting only’ ‘injecting and non injecting’, ‘not specified’ etc. In the results text you should mention the proportion of studies that included only injecting drug users. Any other type of information about what kind of drug users the patient cohorts comprise of will also improve the precision of your meta-analysis and provide a better framework for the design of future meta-analyses.

·      Remove the Table column entitled ‘observational study’ and mention this was the type of study for all of the included original research, in your methods or results section, and also as a footnote in the table

·      After addressing the above points, please also expand on your comments after line 224 in the discussion to comment on how your meta-analysis has (or has not) provided further information about infection rates/risks of this new emerging population of non-injecting drug users, and provide an opinion about whether you recommend the design of future meta-analyses in relation, and or how one could improve the design of future meta-analyses

·      Please revise the discussion section to include the word ‘injecting’ before the term ‘drug users’, wherever this would improve the accuracy of the phrase.

·      Please include a more detailed Figure legend for all of the figures because they are very poorly developed or non-existent at the moment. More detail is especially needed to define all of the abbreviations used, the meaning of axes, more precise figure legend titles, and an explanation of the type of statistics used. Some statistics from the figure could optionally be moved to the legend, when appropriate, to reduce the complexity of the figure.

·      Please remove the sections titled Appendix A and B if there is no additional information for these sections.

·      Please update the acknowledgements section

·      The pixel resolution of the figures could be improved for better visualization of the results

Additional recommendations:

·      If only studies including injecting drug users were used please add the word ‘injecting’ to the article title to more precisely describe the type of drug users that have been investigated, i.e., ‘…infection among injecting drug users – findings from…’

Reviewer 2 Report

Hepatitis C is a viral infectious disease caused by hepatitis C virus (HCV). This paper is mainly focused on determining the factors correlating with the HCV infection. Patients data from 49 studies were used to analyze the correlation between HCV infection and 7 different factors, such as the injection and duration of the drug use, marriage status, sexual behaviors, and etc. Through this meta-analysis, the results indicate the sterile needles and syringes, reducing or safe drug-use behaviors, and increasing the methadone treatment will reduce the HCV infection.

Major concerns

1.     In general, the conclusions from this paper have little novelty. The datasets come from the previous publications, and the authors did not add any more information in addition to the known conclusions. It is more like a review paper than a research paper.

2.     The study made use of the data from 49 different papers, there must be so many confounding factors. The authors did not evaluate and remove all these confounding factors as well, which make the conclusions less solid.

3.     In general, the results haven’t been very well demonstrated in the result section. Line 172, the authors mentioned: “These results are detailed in Figures 2, 3 and 4.” Instead of asking audience to read the plots by themselves, the authors are supposed to describe the results in the manuscript and draw the conclusions in this section.

Reviewer 3 Report

The results are not surprising but it is useful to reinforce this message.

The authors should mention something about the new curative DAA (directly acting antiviral) HCV treatments which are now pan-genotypic - to keep this article topical and up-to-date.

These articles might be a good start: 

https://www.ncbi.nlm.nih.gov/pubmed/28319996 

https://www.ncbi.nlm.nih.gov/pmc/articles/PMC5544136/ 

https://www.cochranelibrary.com/cdsr/doi/10.1002/14651858.CD012143.pub3/media/CDSR/CD012143/CD012143.pdf